# Effect of Pravastatin and Simvastatin on the Reduction of Cytochrome C

**DOI:** 10.3390/jpm12071121

**Published:** 2022-07-10

**Authors:** Krisztián Csomó, Andrea Belik, András Hrabák, Benedek Kovács, Orsolya Fábián, Sándor Valent, Gábor Varga, Zoltán Kukor

**Affiliations:** 1Department of Molecular Biology, Faculty of Medicine, Semmelweis University, Tűzoltó utca 37-47, 1094 Budapest, Hungary; csomo.krisztian@dent.semmelweis-univ.hu (K.C.); belik.andrea@phd.semmelweis.hu (A.B.); hrabak.andras@med.semmelweis-univ.hu (A.H.); kovacsgezabenedek@gmail.com (B.K.); orsolyafabian@gmail.com (O.F.); 2Department of Conservative Dentistry, Faculty of Dentistry, Semmelweis University, Szentkirályi utca 47, 1088 Budapest, Hungary; 3Department of Obstetrics and Gynecology, Faculty of Medicine, Semmelweis University, Üllői út 78/A, 1082 Budapest, Hungary; valent.sandor@gmail.com; 4Department of Oral Biology, Faculty of Dentistry, Semmelweis University, Nagyvárad tér 4, 1089 Budapest, Hungary; varga.gabor@dent.semmelweis-univ.hu

**Keywords:** statins, apoptosis, cytochrome c, mitochondrial oxygen consumption, hypercholesterolemia, preeclampsia

## Abstract

Statins are used to treat hypercholesterolemia, with several pleiotropic effects. Alongside their positive effects (for example, decreasing blood pressure), they can also bring about negative effects/symptoms (such as myopathy). Their main mechanism of action is inducing apoptosis, the key step being the release of cytochrome c from the mitochondria. This can be facilitated by oxidative stress, through which glutathione is oxidized. In this research, glutathione was used as a respiratory substrate to measure the mitochondrial oxygen consumption of rat liver with an O_2_ electrode. The reduction of cytochrome c was monitored photometrically. Hydrophilic (pravastatin) and lipophilic (simvastatin) statins were used for the measurements. Pravastatin reduces the reduction of cytochrome c and the oxygen consumption of the mitochondria, while simvastatin, on the other hand, increases the reduction of cytochrome c and the mitochondrial oxygen consumption. The results make it seem probable that statins influence the mitochondrial oxygen consumption through cytochrome c. Simvastatin could enhance the oxidizing capacity of free cytochrome c, thereby increasing oxidative stress and thus facilitating apoptosis. The observed effects could further the understanding of the mechanism of action of statins and thereby aid in constructing optimal statin therapy for every patient.

## 1. Introduction

Statins are pharmaceutical drugs used primarily to treat hypercholesterolemia, lowering cholesterol synthesis by blocking the enzyme HMG-CoA reductase. Apart from their specific inhibitor effect, statins have several targets/points of effect and thereby bring about a pleiotropic effect. Among these are several useful effects, such as the reduction of VLDL and LDL levels, the slight lowering of blood pressure, anti-inflammatory effect, neuroprotective effect, and antitumor effect [1]. On the other hand, statins have several negative effects as well, including the rather commonly occurring rhabdomyolysis, myopathy, and acute pancreatitis. Some statins, for example, simvastatin, induce apoptosis although the exact underlying mechanism is not entirely clear. It has been observed that cytochrome c release increases in cases of simvastatin-induced apoptosis [2]. Free cytochrome c can oxidize glutathione, thereby aiding the release of cytochrome c from the mitochondria. The reduction of cytochrome c is aided by increasing the pH (to pH 8) as well as high levels of Mg^2+^ and/or Ca^2+^ [3]. Mg^2+^ supplementation usually has beneficial effects though it has been observed in human placenta cultures that a mM concentration of Mg^2+^ actually enhances apoptosis of the cells [4]. Under physiological conditions, cyt c has a net charge of +8 from its unevenly distributed ionizable groups. This favors interactions with negatively charged molecules, such as the polar head of phospholipids, including cardiolipin [2]. This is provided through ionic and apolar bonds. Membrane-bound cytochrome c can only accept electrons from complex III, which it then passes on to complex IV for the reduction of molecular oxygen [5].

Cardiolipin-bound cyt c may also have peroxidase activity. Cardiolipin oxidation by cyt c at the onset of apoptosis is a decisive step. Cyt c is a key Janus catalyst of cardiolipin signaling rather than simply a passive messenger. The peroxidase activity of cyt c can exert a protective role in mitochondria under certain conditions [2]. The reduction of lipoid hydroperoxide compounds to hydroxyl ones provides a way of relieving oxidative stress in the mitochondrial membrane while at the same time generating signaling molecules [6].

Statins can be divided in to lipophilic (simvastatin) and hydrophilic (pravastatin) groups depending on their solubility. Lipophilic and hydrophilic statins can have different side-effects [7,8,9,10]. Based on the information above, it is possible that statins can directly influence the activity of cytochrome c. The experiments detailed below were performed using pravastatin from the hydrophilic group and simvastatin from the lipophilic group.

## 2. Materials and Methods

Simvastatin, cytochrome c, glutathione, and cardiolipin were from Sigma-Aldrich Kft. (Budapest, Hungary). Pravastatin was from Calbiochem (La Jolla, CA, USA). Other chemicals were from Reanal (Budapest, Hungary).

Rat livers were obtained from humanely euthanized female Wistar rats (age: 4–5-month-old, average age: 137 days; weight: between 210 and 340 g, average weight: 256 g).

All animal procedures were carried out according to the EU Directive (2010/63/EU) and were approved by the animal ethics committee of the Hungarian National Food Chain Safety Office (PEI/001/2894-11/2014). All applicable institutional and governmental regulations concerning the ethical use of animals were followed during this research.

The diagrams show the speed of reduction of cytochrome c, which is directly proportional to the oxidation speed of glutathione. The oxidation speed of glutathione was not measured directly since a few permille glutathione oxidizes spontaneously during the experiments. The experiments were performed at pH = 8.0, as this is the pH at which the reduction of cytochrome c is achieved at the highest speed. The effects of statins are often analyzed at a concentration of 5–20 μM [11,12,13], but 100 μM concentration is used as well [14]. The reduction of cytochrome c can be measured photometrically using statins at a 100 μM concentration. Both pravastatin and simvastatin were used in 100 μM concentration in order to eliminate any difference in effect due to different concentration.

### 2.1. Measurement of Cytochrome C Reduction

Reduction of cytochrome c was monitored with spectrophotometer (Hitachi U-2001) at 550 nm [3]. The solutions (final volume 100 µL) were prepared in a cuvette at 25 °C, and the reaction was started by the addition of cytochrome c. The sample was homogenized, checked for bubbles, and the measurement was started at 20 s (0 s of measurement). For technical reasons (dilution, homogenization, reaction in progress), the spectrophotometer could not be reset when cytochrome c was added. The measurement lasted 3 min. The initial velocity was calculated from a 60 s absorbance slope by linear regression (ɛ = 21,000 1/mol). Due to lipids, the cuvette was washed twice with distilled water and three times with ethanol after use and then rinsed with distilled water.

### 2.2. Preparation of Mitochondria

Mitochondria were prepared from Wistar rat liver using a standard protocol from Clayton and Shadel [15]. The liver was cut into 1–2 mm slices using a razor blade. Pieces were rinsed twice with homogenization buffer (210 mM mannitol, 70 mM sucrose, 5 mM Tris-HCl (pH 7.5), 1 mM EDTA (pH 7.5)). The liver was added to the homogenization buffer (1:10 = weight/volume) and homogenized using Potter–Elvehjem homogenizer. The supernatant was centrifuged at 1200× *g* for 10 min. The centrifugation was repeated. The supernatant was centrifuged at 11,000× *g* for 15 min to pellet the mitochondria. The mitochondrial pellet was resuspended twice in homogenization buffer and centrifuged at 11,000× *g* for 15 min. The final pellet was resuspended.

A modified biuret method was used to determine mitochondrial protein concentration [16].

### 2.3. Mitochondrial Oxygen Consumption Assay

Respiratory rates were determined by measuring the oxygen consumption of mitochondria using Clark-type electrode in 1 mL sealed chamber, which was stirred at 37 °C [3]. The mitochondria (1 mg protein/mL) were incubated in 250 mM sucrose, 40 mM Tris/HCl (pH 8.0), and 5 mM glutathione. 

IBM SPSS 27.0.1.0 program was used for the statistical analysis. Confidence interval: 95%.

## 3. Results

### 3.1. The Effect of Pravastatin and Simvastatin on the Reduction of Cyt C by GSH

Applying the same circumstances (pH = 8.0; 1 mM EDTA, no Mg^2+^ or Ca^2+^), simvastatin increased reduction speed of cyt c, while pravastatin inhibited reduction speed. The effects of the lipophilic (simvastatin) and the hydrophilic (pravastatin) statins are opposite. The effect is concentration-dependent in both cases. Further experiments showed that the applied 100 μM pravastatin lowered the reduction speed of cyt c (GSH oxidation) by 27%, while 100 μM of simvastatin increased the reduction speed by 36% (Figure 1).

Statistically analyzing the effects of pravastatin and simvastatin in the utilized concentrations on the reduction of cytochrome c showed significant difference using the independent samples Kruskal–Wallis test.

Based on the Games–Howell post hoc test, the difference was significant between pravastatin and simvastatin at the following concentrations: 50 μM, 100 μM, and 500 μM (see Figure 1).

The presence of Mg^2+^ increased the reduction rate of cyt c using pravastatin as well as simvastatin. Mg^2+^ of 1 mM concentration decreased the cytochrome c reduction with pravastatin to 17%, while the simvastatin-induced cytochrome c reduction speed increased slightly at pH = 8.0 (Figure 2).

Statistically, in the presence of 1 mM Mg^2+^, 100 μM of pravastatin and 100 μM simvastatin affected the reduction of mitochondrial cytochrome c in opposite ways. This difference is significant based on the independent samples Kruskal–Wallis test (*p* = 0.007). Based on the Games–Howell post hoc, test the difference was significant between pravastatin and simvastatin (see Figure 2).

In the presence of cardiolipin (200 μg/mL), the GSH oxidizing capacity of cyt c was reduced. Neither 100 μM pravastatin nor 100 μM simvastatin had a significant influence on the speed of oxidation in the presence of cardiolipin. The joint presence of Mg^2+^ (1.0 mM) and cardiolipin (200 μg/mL) significantly (*p* < 0.05) increased the oxidation speed of glutathione. This can be explained by the fact that Mg^2+^ can bind to either cardiolipin or cyt c and thereby freeing cytochrome c, which in turn oxidizes glutathione. Further, 100 μM pravastatin decreased the oxidation of GSH by cyt c by 15% in the presence of Mg^2+^ and CL, while 100 μM simvastatin increased the reduction by 24% compared to the Mg^2+^ plus CL control (Figure 3).

### 3.2. The Effect of Pravastatin and Simvastatin on the Oxygen Consumption of Rat Liver

Rat liver mitochondria can also use GSH as respiratory substrate [3]. The rate of oxygen consumption increased under the influence of Mg^2+^ until it reached a concentration of 1 mM, while the oxygen consumption rate started to decrease at a 10 mM Mg^2+^ concentration.

In the presence of 1 mM Mg^2+^, the oxygen consumption of the mitochondria decreased slightly by 12% when using pravastatin, while the use of simvastatin increased oxygen consumption by 22% (Figure 4).

In the presence of 1 mM Mg^2+^, 100 μM of pravastatin and 100 μM simvastatin affected the mitochondrial oxygen consumption in opposite ways. This difference is significant based on the independent samples Kruskal–Wallis test (*p* = 0.024). Based on the Games–Howell post hoc test, the difference was significant between the effect of pravastatin and simvastatin (see Figure 4).

## 4. Discussion

Statins can influence apoptosis in different ways. The results are equivocal though lipophilic statins rather facilitate while hydrophilic statins on the other hand either do not influence or they impede the apoptotic processes [13]. Lipophilic and hydrophilic statins have significantly different properties. Only lipophilic statins (simvastatin, atorvastatin, lovastatin, fluvastatin, cerivastatin, pitavastatin) have been shown to enhance vascular smooth muscle cells apoptosis even in the presence of survival factors. On the contrary, hydrophilic statins (pravastatin, rosuvastatin) have been reported to inhibit the apoptotic process. Statins promote apoptosis in a variety of ways. They are specific inhibitors of the HMG-CoA reductase, thus reducing the levels of intermediates in cholesterol synthesis (isoprenoids farnesyl pyrophosphate, geranylgeranyl pyrophosphate). The post-translational prenylation of several proteins (Ras, Rho, Rac) is reduced. This regulates a variety of cellular processes, including cellular signaling, differentiation, and apoptosis. Statin treatment has been shown to inactivate p-21 Rho A protein through inhibition of its prenylation and subsequently downregulate the expression of anti-apoptotic Bcl-2 protein or stimulate the expression of TNFaR, thereby potentiating TNFa-mediated apoptosis. Statins decrease the expression of survival factors such as survivin [17]. In the programmed cell death pathway, cytochrome c has several possible key roles. Simvastatin induces ROS formation in KKU-100 cells but not in KKU-M214 cells (both are cholangiocarcinoma cell line). Simvastatin enhanced the release of cytochrome c, caspase 3, and increased p21 levels in cholangiocarcinoma cell line, especially for the KKU-100 cells [18]. 

Differences in effect between lipophilic and hydrophilic statins have been observed examining gynecological cancers (ovarian, endometrial, and cervical) cell cultures. The lipophilic statins (lovastatin and simvastatin) activated the extrinsic and intrinsic apoptotic cascade and facilitated the emission of cytochrome c from the mitochondria. On the other hand, pravastatin did not induce apoptosis. It is noteworthy that the cancerous cells were the ones receptive to the apoptotical effects of the statins compared to the healthy cells [19]. 

When the cell death program begins, and cyt c leaves its place in the respiratory chain as an electron carrier, it severs the link between complex III and IV. This leads to a decrease in the mitochondrial membrane potential as well as a decrease in the rate of ATP synthesis. This in turn leads to an increased generation of ROS in the mitochondria, which may induce or sustain the cell death program. Cytochrome c also has further roles in apoptosis, as it activates the caspase cascade (as a part of the apoptosoma). Cytochrome c therefore has an important role in cells, as it helps maintain the redox balance of the cell, and it can act as an antioxidant [20]. 

Apoptosis is an important part of cell regulation and needed for the organism to eliminate unwanted cells. Statins can have a beneficial effect in this process, more precisely as a side effect: their antitumor effect. This antitumor effect is multifactorial, and the following contribute: oxidative stress, inhibition of proliferation and metastasis, cell cycle arrest, and induction of apoptosis [21]. 

When examining the antitumor effect of statins, researchers found significant differences between hydrophilic and lipophilic statin users. Meta-analyses of Wang et al. and Li et al. showed that lipophilic statins can help prevent hepatocellular carcinoma. Hydrophilic statins on the other hand cannot. This association can mostly be seen in patients with the highest accumulative dose of statins compared to the patients with the lowest accumulative dose [22,23]. The antitumor effect of statins has been observed by others as well. Liu et al. examined statin use and breast cancer rates and found similar results. While lipophilic statins showed a strong protective function in breast cancer patients, hydrophilic statins on the other hand only showed slightly improved overall mortality [24]. 

A simvastatin-loaded cubosoma may cause ferroptosis and apoptosis in breast cancer cells. Decreased glutathione and increased reactive oxygen species (ROS) levels were observed. Due to the increased ROS level, the concentration of glutathione may be decreased by glutathione peroxidase, but significantly decreased glutathione peroxidase 4 levels have been observed [25]. This discrepancy can be resolved by our observation that free cytochrome c can decrease reduced glutathione concentration independently of peroxidase. 

Fluvastatin (lipophilic statin) can induce the mitochondrial pathway of apoptosis in hepatocellular carcinoma cell lines [22]. During this process, the expression of anti-apoptotic Bcl2 diminishes, while the expression of pro-apoptotic Bax increases. Cytochrome c exiting into the cytosol can be observed as well as caspase activation [3]. Kaufmann et al. examined the effects of lipophilic statins and hydrophilic pravastatin on rat skeletal muscle cell line and on mitochondria. At a statin concentration of 100 μM, lipophilic statins induced apoptosis and facilitated the emission of cytochrome c from the mitochondria, whereas pravastatin on the other hand did not, and they inhibited the β-oxidation of fatty acids considerably more than pravastatin. As already demonstrated, statins do cause a loss of the mitochondrial membrane potential [26].

It is well-known that free cytochrome c can oxidize superoxide to oxygen, thus having an antioxidant and antiapoptotic effect. The absorbed electron can be transferred to the complex IV by binding to the cytochrome c membrane. If cytochrome c is reduced by glutathione rather than superoxide, cytochrome c loses its antioxidant properties. Simvastatin may potentiate the apoptotic pathway by increasing the oxidative properties of cytochrome c glutathione. Hancock et al. hypothesized that the apoptotic effect of cytochrome c may also depend on the state of redox [20]. Accordingly, the oxidized free cytochrome c has a stronger apoptotic effect than the reduced form [27]. This is a hypothesis that may contradict our measurements. Reduction of cytochrome c with glutathione may have antiapoptotic effects. According to our measurements, the oxygen consumption of mitochondria is increased by simvastatin, suggesting that cytochrome c may be reoxidized, and the electron of glutathione will eventually be converted to molecular oxygen. Hancock’s hypothesis is contradicted by the observation that the apoptotic effects of cytochrome c are independent of the redox state [28]. 

This work can help specify the differences in the mechanism of effect between lipophilic and hydrophilic statins. During physiological circumstances, pravastatin and simvastatin binds to cardiolipin and hence do not influence the glutathione oxidating capacity of cytochrome c. Dissociating from the membrane, pravastatin reduced the reduction of cytochrome c, while simvastatin increased it. From these findings, it can be deduced that pravastatin and simvastatin alter the oxygen consumption of the mitochondria by influencing the oxidating capacity of cytochrome c.

When using glutathione as a respiratory substrate, the level of oxygen consumption cannot be altered significantly using either ADP or dinitrophenol. This means that the oxidation of glutathione does not result in ATP synthesis, and oxygen consumption is independent of ATP synthesis.

Based on these results, the following model was set up (Figure 5). In the case of intact mitochondria, statins do not influence the functioning of cytochrome c bound to the inner membrane. If cytochrome c passes into the intermembrane space, it can oxidate glutathione and transport the accepted electron directly to complex IV, and thereby glutathione can increase the oxygen consumption of the mitochondria. The oxidation of glutathione is independent of the proton gradient in contrast to the oxidation of the physiological respiratory substrates. Simvastatin can facilitate while pravastatin can obstruct the oxidating effect of cytochrome c. The oxidation of glutathione facilitates the emission of cytochrome c from the mitochondria. Therefore, in the case of mitochondrial damage, simvastatin could aid apoptosis, while pravastatin could impede it.

The possible therapeutic uses based on the pleiotropic effects of statins are currently intensely studied [29,30]. The diversity of the effects of statins could open up entirely different therapeutic possibilities. Based on our results, it is worth reconsidering the use of statins if the therapy involves larger quantities of Mg^2+^ (for example preeclampsia), especially if increased apoptosis is observed. In the case of subarachnoid hemorrhage, the death toll at 14 days almost doubled with the combined use of statins and Mg^2+^ compared to statin-only therapy. The apoptosis-inducing and cell cycle-inhibiting effect of simvastatin can also be useful. It is known that simvastatin increases the apoptosis of cancerous cells [3,4,5]. Based on our results, it is worth considering supplementing the simvastatin therapy of cancerous cells with Mg^2+^ in order to aid the apoptotic processes. The apoptotic process can also be facilitated by cytochrome c oxidizing glutathione, thereby lowering the ATP level since the use of oxygen and ATP synthesis is dissociated.

Preeclampsia is one of the most dangerous disorders of pregnancy, and there is still no definite cure. Statin therapy is one of several promising therapies. The inhibition of cell division and facilitation of apoptosis during pregnancy is harmful; therefore, in these cases, hydrophilic statin (pravastatin) therapy is preferred over lipophilic statins. The use of pravastatin is further supported by the fact that pravastatin could facilitate the uptake of arginine by the placenta, which can be useful in arginine-deficient preeclampsia [31,32].

The widespread use of statins and their diverse mechanism of action, especially concerning their effect on apoptosis, necessitates further, more specific studies in clinical settings as well, for which this paper can serve as a basis.

## Figures and Tables

**Figure 1 jpm-12-01121-f001:**
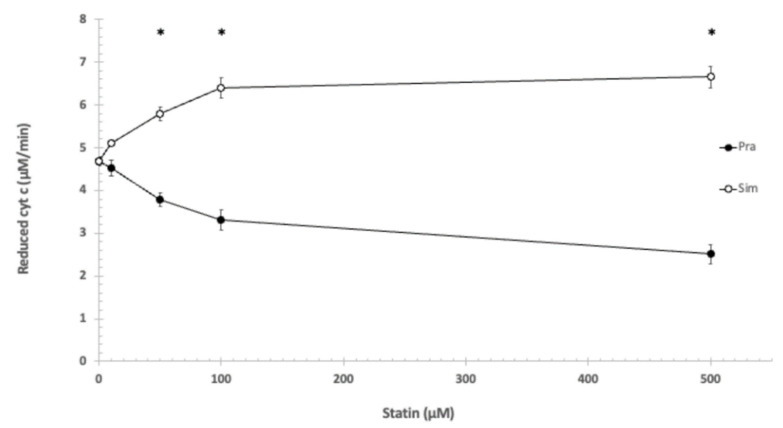
The effect of pravastatin and simvastatin on the reduction of cytochrome c. 1 mM EDTA, 100 mM Tris/HCl (pH = 8.0), T = 25 °C. * shows significant difference.

**Figure 2 jpm-12-01121-f002:**
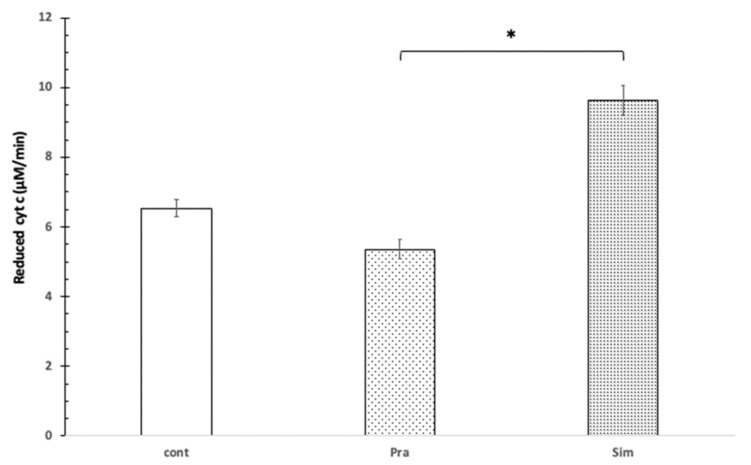
The effect of pravastatin and simvastatin on cytochrome c reduction in the presence of Mg^2+^. 100 μM cytochrome c, control without pravastatin and simvastatin (cont), 100 μM pravastatin (Pra), 100 μM simvastatin (Sim), 100 mM Tris/HCl (pH = 8.0), 1 mM Mg^2+^, T = 25 °C. * shows significant difference.

**Figure 3 jpm-12-01121-f003:**
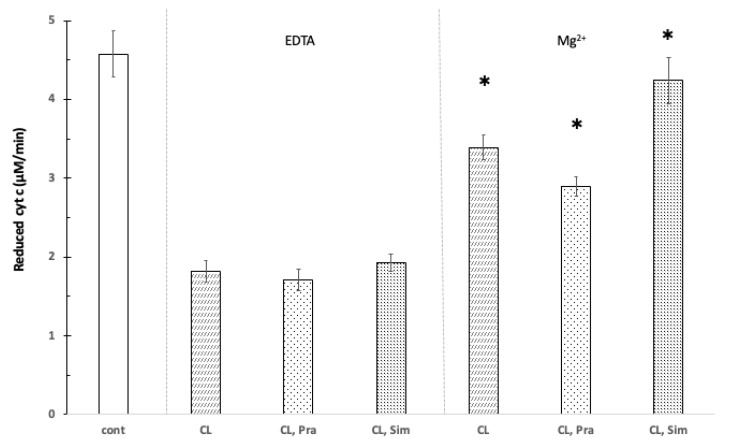
The effect of pravastatin and simvastatin on the reduction of cytochrome c in the presence of cardiolipin. Control without cardiolipin, pravastatin, and simvastatin (cont); 200 μg/mL cardiolipin (CL), 100 μM cytochrome c, 100 μM pravastatin (Pra), 100 μM simvastatin (Sim), 100 mM Tris/HCl (pH 8.0), 1 mM Mg^2+^ (Mg), 1 mM EDTA (EDTA), T = 25 °C. * shows significant difference.

**Figure 4 jpm-12-01121-f004:**
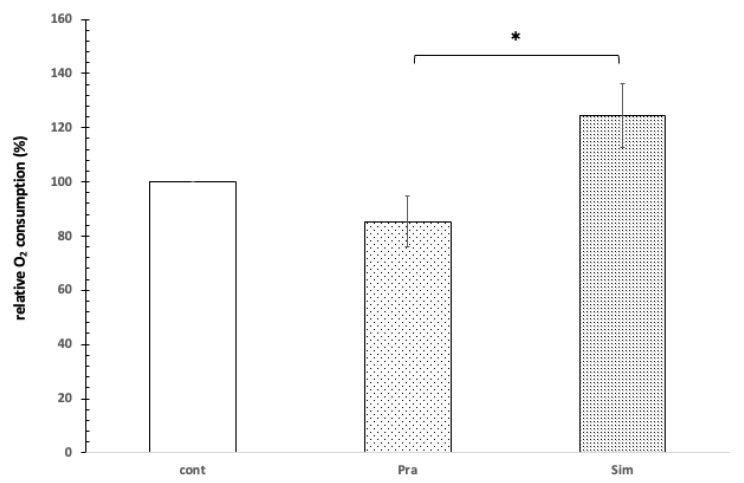
The effect of pravastatin and simvastatin on the oxygen consumption of rat liver mitochondria. Control without pravastatin and simvastatin (cont), 100 μM pravastatin (Pra), 100 μM simvastatin (Sim), 1 mM Mg^2+^, 1 mM GSH, 1 mg protein/mL, pH = 8.0, T = 37 °C. * shows significant difference.

**Figure 5 jpm-12-01121-f005:**
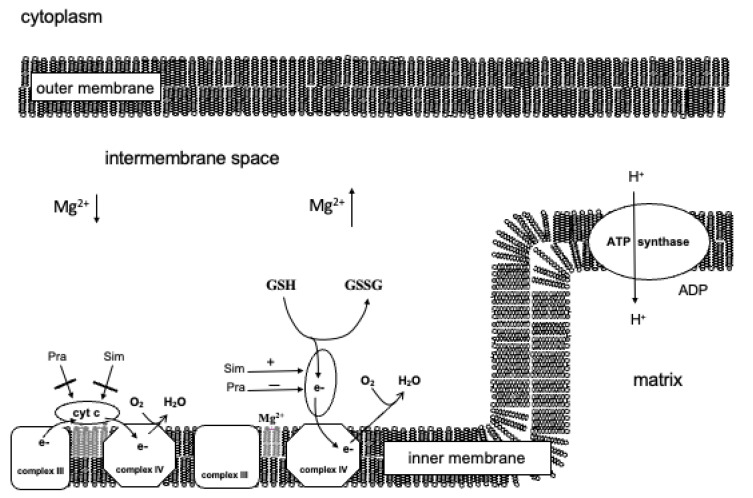
The assumed direct effect of pravastatin and simvastatin on the mitochondrion.

## Data Availability

Not applicable.

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
