# Peer review of "Effect of Pravastatin and Simvastatin on the Reduction of Cytochrome C"

_jpm, 2022, doi:10.3390/jpm12071121_

Round 1
Reviewer 1 Report
In this manuscript the authors study the influence of two specific statins, namely Pravastatin (a hydrophilic statin) and Simvastatin (a lipophilic statin), on the reduction of cytochrome C, a key step in apoptosis (cell death). While statins have dominated in effectively treating hypercholesterolemia for many years, unfortunately they also have several side-effects of concern, including apoptosis. The authors have undertaken the work to study how different statins can affect apoptosis. Therefore, without question, the clinical significance of their study is very high. However, this reviewer has the following concerns.
1. The manuscript is not very well written. It lacks clarity and accuracy in many places. It lacks the flow to readers. For example: a) on lines 81 and 82 they have not made it clear whether “oxidation” they are referring to is the oxidation of glutathione. The subtitle of 3.1. refers to reduction of Cytochrome C. So, it is more appropriate to refer to reduction of cytochrome c rather than oxidation of presumably glutathione. b) On line 72, should it not be the reduction of cytochrome c by glutathione instead of oxidation? c) on lines 98 and 99 it is not clear if they are talking about the protective efficacy of pravastatin against oxidation or about oxidation of pravastatin. d) In figures 3 and 4 I assume “cont” means “control” but it is not defined anywhere and also no one knows what this control is composed of. e) on lines 115 and 116 it says pravastatin decreases the reduction of GSH, should it not be oxidation of GSH?
2. The authors do not provide sufficient experimental details in Materials and Methods section. For example, a) on line 74 it is mentioned that the reduction of cytochrome c can be measured photometrically, but did they actually use this methodology and if so where are details? b) there is no experimental description of how they measured oxygen consumption in Materials and Methods section and also in Figure 4 legend.
3. Figure 1-3 results are obtained by just mixing chemicals, but not much experimental details are provided such as quantities used and time of reaction and any other conditions. Is there any evidence that the results obtained reflect the biological processes occurring in vivo?
4. There is no Figure 6 in this manuscript (line 168). Please correct it to Figure 5.
Author Response
Open Review 1
Dear Reviewer,
Thank you for your work with our paper. We have made the suggested corrections and we have answered your questions, please see our detailed answers below. We left out the references here in the answers, but they are included in the corrected paper.
In this manuscript the authors study the influence of two specific statins, namely Pravastatin (a hydrophilic statin) and Simvastatin (a lipophilic statin), on the reduction of cytochrome C, a key step in apoptosis (cell death). While statins have dominated in effectively treating hypercholesterolemia for many years, unfortunately they also have several side-effects of concern, including apoptosis. The authors have undertaken the work to study how different statins can affect apoptosis. Therefore, without question, the clinical significance of their study is very high. However, this reviewer has the following concerns.
- The manuscript is not very well written. It lacks clarity and accuracy in many places. It lacks the flow to readers. For example: a) on lines 81 and 82 they have not made it clear whether “oxidation” they are referring to is the oxidation of glutathione. The subtitle of 3.1. refers to reduction of Cytochrome C. So, it is more appropriate to refer to reduction of cytochrome c rather than oxidation of presumably glutathione. b) On line 72, should it not be the reduction of cytochrome c by glutathione instead of oxidation? c) on lines 98 and 99 it is not clear if they are talking about the protective efficacy of pravastatin against oxidation or about oxidation of pravastatin. d) In figures 3 and 4 I assume “cont” means “control” but it is not defined anywhere and also no one knows what this control is composed of. e) on lines 115 and 116 it says pravastatin decreases the reduction of GSH, should it not be oxidation of GSH?
- corrected: Applying the same circumstances (pH=8.0; 1 mM EDTA, no Mg2+ or Ca2+), simvastatin increases reduction speed of Cyt c while pravastatin inhibits reduction
- corrected: The experiments were performed at pH=8.0, as this is the pH at which the reduction of cytochrome c is achieved at the highest speed.
- corrected: Mg2+ of 1 mM concentration decreased the cytochrome c reduction with pravastatin to 17%, while the simvastatin-induced cytochrome c reduction speed increased slightly at pH=8.0 (Figure 2.).
- corrected: Figure 2. Control without pravastatin and simvastatin (cont)
Figure 3. Control without cardiolipin, pravastatin and simvastatin (cont)
Figure 4. Control without pravastatin and simvastatin (cont)
- corrected: 100 mM pravastatin decreases the oxidation of GSH by cyt c by 15% in the presence of Mg2+ and CL, while 100 mM simvastatin increases the reduction by 24% compared to the Mg2+, CL control (Figure 3.).
- The authors do not provide sufficient experimental details in Materials and Methods section. For example, a) on line 74 it is mentioned that the reduction of cytochrome c can be measured photometrically, but did they actually use this methodology and if so where are details? b) there is no experimental description of how they measured oxygen consumption in Materials and Methods section and also in Figure 4 legend.
- and b) We supplemented the Methods.
2.1. Measurement of cytochrome c reduction
Reduction of cytochrome c was monitored with spectrophotometer (Hitachi U-2001) at 550 nm [3]. The solutions (final volume 100 µl) were prepared in a cuvette at 25 °C and the reaction was started by the addition of cytochrome c. The sample was homogenized, checked for bubbles, and the measurement was started at 20 seconds (this is 0 seconds of measurement). For technical reasons (dilution, homogenization, reaction in progress), the spectrophotometer could not be reset when cytochrome c was added. The measurement lasted 3 minutes. The initial velocity was calculated from a 60-second absorbance slope by linear regression (É› = 21,000 1/mol). Due to lipids, the cuvette was washed twice with distilled water and three times with ethanol after use, and then rinsed with distilled water.
2.2. Preparation of mitochondria
Mitochondria were prepared from Wistar rat liver using a standard protocol from Clayton and Shadel [15]. The liver was cut into 1-2-mm slices using a razor blade. Pieces were rinsed twice with homogenization buffer (210 mM mannitol, 70 mM sucrose, 5 mM Tris-HCl (pH 7.5), 1 mM EDTA (pH 7.5). The liver was added to the homogenization buffer (1:10=weight/volume) and homogenized using Potter-Elvehjem homogenizer. The supernatant was centrifuged at 1200 g for 10 min. Centrifugation was repeated. The supernatant was centrifuged at 11,000 g for 15 min to pellet the mitochondria. The mitochondrial pellet was resuspended twice in homogenization buffer and centrifuged at 11,000 g for 15 min. The final pellet was resuspended.
A modified biuret method was used to determine mitochondrial protein concentration [16].
2.3. Mitochondrial oxygen consumption assay
Respiratory rates were determined by measuring the oxygen consumption of mitochondria using Clark type electrode in 1 ml sealed chamber, which was stirred at 37 °C [3]. The mitochondria (1 mg protein/ml) were incubated in 250 mM sucrose, 40 mM Tris/HCl (pH 8.0), 5 mM glutathione.
- Figure 1-3 results are obtained by just mixing chemicals, but not much experimental details are provided such as quantities used and time of reaction and any other conditions. Is there any evidence that the results obtained reflect the biological processes occurring in vivo?
A simvastatin-loaded cubosoma may cause ferroptosis and apoptosis in breast cancer cells. Decreased glutathione and increased reactive oxygen species (ROS) levels were observed. Due to the increased ROS level, the concentration of glutathione may be decreased by glutathione peroxidase, but significantly decreased glutathione peroxidase 4 levels have been observed [25]. This discrepancy can be resolved by our observation that free cytochrome c can decrease reduced glutathione concentration independently of peroxidase.
It is well known that free cytochrome c can oxidize superoxide to oxygen, thus having an antioxidant, antiapoptotic effect. The absorbed electron can be transferred to the complex IV by binding to the cytochrome c membrane. If cytochrome c is reduced by glutathione rather than superoxide, cytochrome c loses its antioxidant properties. Simvastatin may potentiate the apoptotic pathway by increasing the oxidative properties of cytochrome c glutathione. Hancock et al. hypothesized that the apoptotic effect of cytochrome c may also depend on the state of redox [20]. Accordingly, the oxidized free cytochrome c has a stronger apoptotic effect than the reduced form [27]. This is a hypothesis that may contradict our measurements. Reduction of cytochrome c with glutathione may have antiapoptotic effects. According to our measurements, the oxygen consumption of mitochondria is increased by simvastatin, suggesting that cytochrome c may be reoxidized and the electron of glutathione will eventually be converted to molecular oxygen. Hancock’s hypothesis is contradicted by the observation that the apoptotic effects of cytochrome c are independent of the redox state [28].
- There is no Figure 6 in this manuscript (line 168). Please correct it to Figure 5.
Corrected.
Reviewer 2 Report
Dear authors,
please consider the following recommendations when editing your article. In my view, there are several shortcomings in this article, so I do not recommend it for publication in its current form.
In the introduction, the characteristics of cytochrome c are described chaotically, first, you describe the effect of statins on apoptosis, which then triggers the "activation" of cytochrome C, while its role is mainly to be active in mitochondria as an electron shuttle in the respiratory chain. You are describing "free" cytochrome c as a compound capable of oxidizing glutathione, and later that oxidation of glutathione does not lead to the formation of ATP and therefore I do not understand the link between "activation" of cyt c in this sense. Also confusing is the claim that free cyt c is bound to CL under physiological conditions, which is true for stabilizing the structure of the enzyme cyt c oxidase, while it is the CL-bound cyt c that results in peroxidation and subsequent activation of apoptosis (CL-specific peroxidase activity of CL-bound Cyt c).
In the methods, you should indicate the specific concentrations in the methods and not the considerations that should be used. (lines 72-76)
Do the statins affect the reduction (of concentration?) of cyt c "by GSH"? line 80
Figure 1, 2: statin concentration in figure 1: 50-100-500uM - in legend 100uM; reduced cyt c means ferrocytochrome c, is that what you wanted to capture in the chart?
The pathways of statin-induced apoptosis involving cyt c should be discussed in more detail. line 140
You should pay more attention to the description of the advantages and disadvantages of statin-induced apoptosis in specific diseases, as well as to the importance of why you compared lipo- and hydrophilic statins separately, as their therapeutic effect lies precisely in the difference between these pathways.
Author Response
Open Review 2
Dear Reviewer,
Thank you for your work with our paper. We have made the suggested corrections and we have answered your questions, please see our detailed answers below. We left out the references here in the answers, but they are included in the corrected paper.
Dear authors,
please consider the following recommendations when editing your article. In my view, there are several shortcomings in this article, so I do not recommend it for publication in its current form.
In the introduction, the characteristics of cytochrome c are described chaotically, first, you describe the effect of statins on apoptosis, which then triggers the "activation" of cytochrome C, while its role is mainly to be active in mitochondria as an electron shuttle in the respiratory chain. You are describing "free" cytochrome c as a compound capable of oxidizing glutathione, and later that oxidation of glutathione does not lead to the formation of ATP and therefore I do not understand the link between "activation" of cyt c in this sense. Also confusing is the claim that free cyt c is bound to CL under physiological conditions, which is true for stabilizing the structure of the enzyme cyt c oxidase, while it is the CL-bound cyt c that results in peroxidation and subsequent activation of apoptosis (CL-specific peroxidase activity of CL-bound Cyt c).
Corrected:
It has been observed that cytochrome c release increases in cases of simvastatin-induced apoptosis.
Under physiological conditions, Cyt c has a net charge of +8 from its unevenly distributed ionizable groups. This favors interactions with negatively charged molecules, such as the polar head of phospholipids, including cardiolipin [2]. This is provided through ionic and apolar bonds. Membrane-bound cytochrome c can only accept electrons from complex III, which it then passes on to complex IV for the reduction of molecular oxygen [5].
Cardiolipin-bound Cyt c may also have peroxidase activity. Cardiolipin oxidation by Cyt c at the onset of apoptosis is a decisive step. Indeed, Cyt c is a key Janus catalyst of cardiolipin signaling rather than a passive messenger. The peroxidase activity of Cyt c can exert a protective role in mitochondria under certain conditions [2]. Reduction of lipoid hydroperoxide compounds to hydroxyl ones provides a way of relieving oxidative stress in the mitochondrial membrane while generating signaling molecules [6].
In the methods, you should indicate the specific concentrations in the methods and not the considerations that should be used. (lines 72-76)
Corrected:
2.1. Measurement of cytochrome c reduction
Reduction of cytochrome c was monitored with spectrophotometer (Hitachi U-2001) at 550 nm [3]. The solutions (final volume 100 µl) were prepared in a cuvette at 25 °C and the reaction was started by the addition of cytochrome c. The sample was homogenized, checked for bubbles, and the measurement was started at 20 seconds (this is 0 seconds of measurement). For technical reasons (dilution, homogenization, reaction in progress), the spectrophotometer could not be reset when cytochrome c was added. The measurement lasted 3 minutes. The initial velocity was calculated from a 60-second absorbance slope by linear regression (É› = 21,000 1/mol). Due to lipids, the cuvette was washed twice with distilled water and three times with ethanol after use, and then rinsed with distilled water.
2.2. Preparation of mitochondria
Mitochondria were prepared from Wistar rat liver using a standard protocol from Clayton and Shadel [15]. The liver was cut into 1-2-mm slices using a razor blade. Pieces were rinsed twice with homogenization buffer (210 mM mannitol, 70 mM sucrose, 5 mM Tris-HCl (pH 7.5), 1 mM EDTA (pH 7.5). The liver was added to the homogenization buffer (1:10=weight/volume) and homogenized using Potter-Elvehjem homogenizer. The supernatant was centrifuged at 1200 g for 10 min. Centrifugation was repeated. The supernatant was centrifuged at 11,000 g for 15 min to pellet the mitochondria. The mitochondrial pellet was resuspended twice in homogenization buffer and centrifuged at 11,000 g for 15 min. The final pellet was resuspended.
A modified biuret method was used to determine mitochondrial protein concentration [16].
2.3. Mitochondrial oxygen consumption assay
Respiratory rates were determined by measuring the oxygen consumption of mitochondria using Clark type electrode in 1 ml sealed chamber, which was stirred at 37 °C [3]. The mitochondria (1 mg protein/ml) were incubated in 250 mM sucrose, 40 mM Tris/HCl (pH 8.0), 5 mM glutathione.
Do the statins affect the reduction (of concentration?) of cyt c "by GSH"? line 80
This refers to the redox process (reduction of cytochrome c).
Figure 1, 2: statin concentration in figure 1: 50-100-500uM - in legend 100uM; reduced cyt c means ferrocytochrome c, is that what you wanted to capture in the chart?
Corrected: Figure 2. The effect of pravastatin and simvastatin on cytochrome c reduction in the presence of Mg2+. 100 mM cytochrome c, control without pravastatin and simvastatin (cont), 100 mM pravastatin (Pra), 100 mM simvastatin (Sim), 100 mM Tris/HCl (pH=8.0), 1 mM Mg2+, T=25 oC.
The pathways of statin-induced apoptosis involving cyt c should be discussed in more detail. line 140
Corrected:
Lipophilic and hydrophilic statins have significantly different properties. Only lipophilic statins (simvastatin, atorvastatin, lovastatin, fluvastatin, cerivastatin, pitavastatin) have been shown to enhance vascular smooth muscle cells apoptosis, even in the presence of survival factors. On the contrary, hydrophilic statins (pravastatin, rosuvastatin) have been reported to inhibit the apoptotic process. Statins promote apoptosis in a variety of ways. They are specific inhibitors of the HMG-CoA reductase, thus reducing the levels of intermediates in cholesterol synthesis (isoprenoids farnesyl pyrophosphate, geranylgeranyl pyrophosphate). The post-translational prenylation of several proteins (Ras, Rho, Rac) is reduced. This regulates a variety of cellular processes, including cellular signaling, differentiation and apoptosis. Statin treatment has been shown to inactivate p-21 Rho A protein through inhibition of its prenylation, and subsequently downregulate the expression of anti-apoptotic Bcl-2 protein or stimulate the expression of TNFaR, thereby potentiating TNFa-mediated apoptosis. Statins decrease the expression of survival factors such as survivin [17]. In the programmed cell death pathway, cytochrome c has several possible key roles. Simvastatin induces ROS formation in KKU-100 cells, but not in KKU-M214 cells (both are cholangiocarcinoma cell line). Simvastatin enhanced the release of cytochrome c, caspase 3, and increased p21 levels in cholangiocarcinoma cell line, especially for the KKU-100 cells [18].
Cyt c left its usual function as an electron carrier in the respiratory chain, thereby severing the link between complex III and complex IV. Therefore, ΔΨm is decreased and the rate of ATP synthesis decreases too. The mitochondrial generation of ROS is also increased, an event that may be important for the induction or sustenance of the cell death program. Cytochrome c has a positive role in the activation of the caspase cascade (part of the apoptosoma). Cytochrome c may have an instrumental role in maintaining the redox balance of the cell and acting as an antioxidant [20].
You should pay more attention to the description of the advantages and disadvantages of statin-induced apoptosis in specific diseases, as well as to the importance of why you compared lipo- and hydrophilic statins separately, as their therapeutic effect lies precisely in the difference between these pathways.
Corrected:
Induction of apoptosis may also have a beneficial effect. The beneficial side effects of statins include their antitumor effects. This is explained by, among other things, induction of apoptosis, cell cycle arrest, oxidative stress and inhibition of proliferation, metastasis [21]. Significant differences were observed between hydrophilic and lipophilic statin users. Meta-analyses of Wang et al and Li et al shows the result that lipophilic statins can prevent hepatocellular carcinoma, while hydrophilic statins cannot. The association was more remarkable in patients with the highest statin accumulative dose compared to those with the lowest accumulative dose [22, 23]. Liu et al drew a similar conclusion when examining the relationship between breast cancer and statin use. Lipophilic statins showed a strong protective function in breast cancer patients, whereas hydrophilic statins only slightly improved all-cause mortality [24].
A simvastatin-loaded cubosoma may cause ferroptosis and apoptosis in breast cancer cells. Decreased glutathione and increased reactive oxygen species (ROS) levels were observed. Due to the increased ROS level, the concentration of glutathione may be decreased by glutathione peroxidase, but significantly decreased glutathione peroxidase 4 levels have been observed [25]. This discrepancy can be resolved by our observation that free cytochrome c can decrease reduced glutathione concentration independently of peroxidase.
It is well known that free cytochrome c can oxidize superoxide to oxygen, thus having an antioxidant, antiapoptotic effect. The absorbed electron can be transferred to the complex IV by binding to the cytochrome c membrane. If cytochrome c is reduced by glutathione rather than superoxide, cytochrome c loses its antioxidant properties. Simvastatin may potentiate the apoptotic pathway by increasing the oxidative properties of cytochrome c glutathione. Hancock et al. hypothesized that the apoptotic effect of cytochrome c may also depend on the state of redox [20]. Accordingly, the oxidized free cytochrome c has a stronger apoptotic effect than the reduced form [27]. This is a hypothesis that may contradict our measurements. Reduction of cytochrome c with glutathione may have antiapoptotic effects. According to our measurements, the oxygen consumption of mitochondria is increased by simvastatin, suggesting that cytochrome c may be reoxidized and the electron of glutathione will eventually be converted to molecular oxygen. Hancock’s hypothesis is contradicted by the observation that the apoptotic effects of cytochrome c are independent of the redox state [28].
Round 2
Reviewer 1 Report
In the revised version of the manuscript authors have satisfactorily answered concerns of this reviewer.
Author Response
Dear Reviewer,
We wish to thank you for your work on our paper and helping us make this article much better. We have made some minor corrections (spell-checking among others) and we will now upload this version.
Best wishes,
the authors
Reviewer 2 Report
Dear authors,
thank you for the extensive corrections you made in your article.
There are still some inconsistent references in the text, which please correct, such as "Cyt c left its usual function as an electron carrier in the respiratory chain, thereby severing the link between complex III and complex IV. Therefore, ΔΨm is decreased and the rate of ATP synthesis decreases too. The mitochondrial generation of ROS is also increased, an event that may be important for the induction or sustenance of the cell death program. - lines 204-207" where it would be appropriate to indicate the conditions under which this dysfunction occurs. It is clear to me that this will be an effect of statins, but it does not sound like it from the text, and in addition, in the next rows you already describe the physiological effects of cyt c.
"Induction of apoptosis may also have a beneficial effect. The beneficial side effects of statins include their antitumor effects. This is explained by, among other things, induction of apoptosis" - vicious circle?
Author Response
Dear Reviewer,
Thank you again for your work with our paper and for taking the time to help improve our article. We have corrected and rewritten the parts you suggested, please see below. We have also made some minor corrections throughout the paper (spell check, sentence structure, formatting the references etc.) that we hope will improve the article overall.
Dear authors,
thank you for the extensive corrections you made in your article.
There are still some inconsistent references in the text, which please correct, such as "Cyt c left its usual function as an electron carrier in the respiratory chain, thereby severing the link between complex III and complex IV. Therefore, ΔΨm is decreased and the rate of ATP synthesis decreases too. The mitochondrial generation of ROS is also increased, an event that may be important for the induction or sustenance of the cell death program. - lines 204-207" where it would be appropriate to indicate the conditions under which this dysfunction occurs. It is clear to me that this will be an effect of statins, but it does not sound like it from the text, and in addition, in the next rows you already describe the physiological effects of cyt c.
Corrected:
When the cell death program begins and cyt c leaves its place in the respiratory chain as an electron carrier, it severs the link between complex III and IV. This leads to a decrease in the mitochondrial membrane potential as well as a decrease in the rate of ATP synthesis. This in turn leads to an increased generation of ROS in the mitochondria, that may induce or sustain the cell death program. Cytochrome c also has further roles in apoptosis, as it activates the caspase cascade (as a part of the apoptosoma). Cytochrome c therefore has an important role in cells as it helps maintain the redox balance of the cell and it can act as an antioxidant [20].
"Induction of apoptosis may also have a beneficial effect. The beneficial side effects of statins include their antitumor effects. This is explained by, among other things, induction of apoptosis" - vicious circle?
Corrected:
Apoptosis is an important part of cell regulation and needed for the organism to eliminate unwanted cells. Statins can have a beneficial effect in this process, more precisely as a side-effect: their antitumor effect. This antitumor effect is multifactorial, and the following contribute: oxidative stress, inhibition of proliferation and metastasis, cell cycle arrest and induction of apoptosis [21].